**Data Availability Statement:** Data cannot be shared publicly because of confidentiality concerns and because it is considered as owned by contributing study sites. Anonymized data are

# Trends in hepatitis C virus coinfection and its cascade of care among adults living with HIV in Asia between 2010 and 2020

Jeremy Ross[1☯]*, Dhanushi Rupasinghe[2☯], Anchalee Avihingsanon[3‡], Man Po Lee[4‡], Sanjay Pujari[5‡], Gerald Sharp[6], Nagalingeswaran Kumarasamy[7‡], Suwimon Khusuwan[8‡], Vohith Khol[9‡], I. Ketut Agus Somia[10‡], Thach Ngoc Pham[11‡], Sasisopin Kiertiburanakul[12‡], Jun Yong Choi[13‡], Cuong Duy Do[14‡], Annette H. Sohn[1‡], Awachana Jiamsakul[2‡], on behalf of the TAHOD-LITE study group of IeDEA Asia-Pacific[¶]

1 TREAT Asia, amfAR, The Foundation for AIDS Research, Bangkok, Thailand, 2 The Kirby Institute, UNSW Sydney, Sydney, NSW, Australia, 3 HIV-NAT/ Thai Red Cross AIDS Research Centre and Tuberculosis Research Unit, Faculty of Medicine, Chulalongkorn University, Bangkok, Thailand, 4 Queen Elizabeth Hospital, Hong Kong SAR, China, 5 Institute of Infectious Diseases, Pune, India, 6 National Institute of Allergy and Infectious Diseases (NIAID), Bethesda, Maryland, United States of America, 7 CART CRS, Voluntary Health Services, Chennai, India, 8 Chiangrai Prachanukroh Hospital, Chiang Rai, Thailand, 9 National Center for HIV/AIDS, Dermatology & STDs, Phnom Penh, Cambodia, 10 Faculty of Medicine, Prof. Dr. I.G.N.G. Ngoerah Hospital, Udayana University, Bali, Indonesia, 11 National Hospital for Tropical Diseases, Hanoi, Vietnam, 12 Faculty of Medicine, Ramathibodi Hospital, Mahidol University, Bangkok, Thailand, 13 Division of Infectious Diseases, Department of Internal Medicine, Yonsei University College of Medicine, Seoul, South Korea, 14 Bach Mai Hospital, Hanoi, Vietnam

☯ These authors contributed equally to this work.
‡ AA, MPL, SP, NK, SK, VK, IKAS, TNP, SK, JYC, CDD, AHS and AJ also contributed equally to this work.
¶ Membership of the TAHOD-LITE study group of IeDEA Asia-Pacific is provided in the Acknowledgments.
* jeremy.ross@treatasia.org

## Abstract

### Background

Chronic hepatitis C virus (HCV) infection contributes to substantial morbidity and mortality among adults living with HIV. Cascades of HCV care support monitoring of program performance, but data from Asia are limited. We assessed regional HCV coinfection and cascade outcomes among adults living with HIV in care from 2010–2020.

### Methods

Patients ≥18 years old with confirmed HIV infection on antiretroviral therapy (ART) at 11 clinical sites in Cambodia, China, India, Indonesia, South Korea, Thailand and Vietnam were included. HCV- and HIV-related treatment and laboratory data were collected from those with a positive HCV antibody (anti-HCV) test after January 2010. An HCV cascade was evaluated, including proportions positive for anti-HCV, tested for HCV RNA or HCV core antigen (HCVcAg), initiated on HCV treatment, and achieved sustained virologic response (SVR). Factors associated with screening uptake, treatment initiation, and treatment response were analyzed using Fine and Gray's competing risk regression model.

available on reasonable request with the agreement of the study and site principal investigators (contact via Anonymized data are available on reasonable request with the agreement of the study and site principal investigators (contact via study project manager tor.peterson@treatasia.org), for researchers who meet the criteria for access to confidential data.

**Funding:** JLR, DR, AA, MPL, SP, NK, SK, VK, IKAS, TNP, SK, JYC, CDD, AHS, AJ received an award to their institutions for The TREAT Asia HIV Observational Database Low-Intensity TransfEr (TAHOD-LITE) study. These awards to their institution were provided through the following grant: IeDEA U01AI069907. The TAHOD-LITE study is an initiative of TREAT Asia, a program of amfAR, The Foundation for AIDS Research, with support from the U.S. National Institutes of Health's (https://www.nih.gov/) National Institute of Allergy and Infectious Diseases, the Eunice Kennedy Shriver National Institute of Child Health and Human Development, the National Cancer Institute, the National Institute of Mental Health, the National Institute on Drug Abuse, the National Heart, Lung, and Blood Institute, the National Institute on Alcohol Abuse and Alcoholism, the National Institute of Diabetes and Digestive and Kidney Diseases, and the Fogarty International Center, as part of the International Epidemiology Databases to Evaluate AIDS (IeDEA; U01AI069907). DR and AJ's institution, The Kirby Institute, is also funded by the Australian Government Department of Health and Ageing, and is affiliated with the Faculty of Medicine, UNSW Sydney. The funders had no role in study design, data collection and analysis, decision to publish, or preparation of the manuscript.

**Competing interests:** AHS has received grants to her institution from ViiV Healthcare and Gilead Sciences. Other authors have declared that no competing interests exist. This does not alter our adherence to PLOS ONE policies on sharing data and materials.

## Results

Of 24,421 patients, 9169 (38%) had an anti-HCV test, and 971 (11%) had a positive result. The proportion with positive anti-HCV was 12.1% in 2010–2014, 3.9% in 2015–2017, and 3.8% in 2018–2020. From 2010 to 2014, 34% with positive anti-HCV had subsequent HCV RNA or HCVcAg testing, 66% initiated HCV treatment, and 83% achieved SVR. From 2015 to 2017, 69% with positive anti-HCV had subsequent HCV RNA or HCVcAg testing, 59% initiated HCV treatment, and 88% achieved SVR. From 2018 to 2020, 80% had subsequent HCV RNA or HCVcAg testing, 61% initiated HCV treatment, and 96% achieved SVR. Having chronic HCV in later calendar years and in high-income countries were associated with increased screening, treatment initiation or achieving SVR. Older age, injecting drug use HIV exposure, lower CD4 and higher HIV RNA were associated with reduced HCV screening or treatment initiation.

## Conclusions

Our analysis identified persistent gaps in the HCV cascade of care, highlighting the need for focused efforts to strengthen chronic HCV screening, treatment initiation, and monitoring among adult PLHIV in the Asia region.

## Introduction

Of the estimated 36.7 million people living with HIV (PLHIV) globally, around 2.3 million have past or current hepatitis C virus (HCV) infection [1]. Worldwide, the overall HCV co-infection prevalence in HIV-positive individuals is 6.2% [2], and chronic HCV infection contributes substantially to the mortality and morbidity of adults living with HIV [3].

In the Asia-Pacific region, the prevalence of HCV co-infection among adult PLHIV ranges from 2.9% to 80% depending on the characteristics of the study population [4–12]. Injecting drug use is consistently associated with an increased risk of HCV coinfection, [10, 13–16] and men who have sex with men (MSM) and sex workers living with HIV are also at increased risk [17, 18]. Other factors associated with HCV coinfection among Asian adult PLHIV include older age, lower education [19], lower CD4 cell count [4, 20], and having a household member with liver disease [21].

HCV coinfection among adult PLHIV in Asia is associated with an increased risk of mortality [22–24]. Liver disease remains one of the most important non-AIDS causes of death among PLHIV and contributes substantial morbidity among Asian adults living with HIV [25, 26], as well as being associated with negative HIV clinical and treatment outcomes, including HIV disease progression, ART attrition and slower CD4 cell count recovery [22, 24, 27, 28].

Cascades of care have been developed to support the monitoring of chronic HCV care and treatment programs and progress towards global targets for HCV elimination as a public health threat by 2030 [29–31]. The vast majority of research on the HCV cascade of care among PLHIV comes from high-income countries (HIC). The limited Asia-Pacific data suggests persistent gaps [32], despite expanding access in recent years to effective direct-acting antiviral (DAA) treatment for chronic HCV.

In order to address these research gaps, we assessed the prevalence of HCV coinfection, trends in the cascade of care for HCV, and factors associated with poor HCV outcomes among adult PLHIV in Asia between 2010 and 2020, using data from the TREAT Asia HIV

Observational Database- Low Intensity TransfEr (TAHOD-LITE) cohort. Assessment of these trends, identification of continuing cascade gaps, and of risk factors for suboptimal cascade outcomes may inform the delivery of comprehensive HCV services to adult PLHIV in the region, and support the reduction of HCV-related morbidity and mortality.

## Methods

### Study design and study population

TAHOD-LITE is a prospective observational cohort of the International Epidemiology Databases to Evaluate AIDS (IeDEA) Asia-Pacific, whose methods have previously been described [33]. Patients currently or previously attending participating clinics, with documented confirmed HIV infection, and 18 years of age or older at enrolment were eligible for inclusion. Data transfers of a core set of patient demographic, hepatitis serology, HIV immunology and virology, antiretroviral (ART) treatment, and limited laboratory data (glucose, creatinine), occurred approximately every two years between 2014 to 2021. The 2021 data transfer included data from over 55,000 adult PLHIV at 11 HIV clinical sites in Cambodia (National Center for HIV/AIDS, Dermatology & STDs), Hong Kong SAR (Queen Elizabeth Hospital), India (Voluntary Health Services; Institute of Infectious Diseases), Indonesia (Sanglah Hospital), South Korea (Severance Hospital), Thailand (Chiangrai Prachanukroh Hospital; HIV-NAT Research Collaboration; Ramathibodi Hospital) and Vietnam (Bach Mai Hospital; National Hospital of Tropical Diseases).

In addition to the core set of data, the 2021 TAHOD-LITE data transfer included more detailed hepatitis C serology, treatment, complications, laboratory and monitoring data from study participants with a positive hepatitis C virus antibody (anti-HCV) result from 1st January 2010 onwards. In our study, we included patients who were seen at the participating clinic from 2010 onwards and had started any antiretroviral therapy (ART). Our cohort had complete follow-up data available to the 31st December 2020, so participants seen after this date had their follow-up censored on this date.

### Statistical analyses

The proportion of PLHIV with a positive anti-HCV test was reported over three time periods that broadly correspond to very limited DAA access (the pre-DAA era), initial DAA access, and expanded DAA access in the Asia-Pacific region: 2010–2014, 2015–2017, and 2018–2020. The HCV cascade of care was defined and evaluated as the proportions (i) tested positive for anti-HCV, (ii) tested for HCV viral load (HCV RNA) or HCV core antigen (HCVcAg), (iii) tested positive for HCV RNA or HCVcAg, (iv) initiated on HCV treatment, (v) completed HCV treatment, (vi) had post-treatment HCV RNA test, and (vii) achieved/reached sustained virologic response (SVR). Factors associated with HCV cascade of care outcomes were assessed focusing on uptake of screening for HCV co-infection, HCV treatment initiation, and HCV treatment response separately.

Screening for chronic HCV co-infection was defined as receiving an anti-HCV test from 2010 onwards. Patients were followed from the later date of cohort entry, start of ART, or 1st January 2010 when viral hepatitis variables were collected in the cohort. Follow-up for these patients ended on the date of anti-HCV screening test for those tested and on the censor date of 31st December 2020 for those never tested. A Kaplan-Meier plot of time from HCV diagnosis to treatment initiation by time period assessed the probability of not initiating HCV treatment. HCV treatment initiation was defined as having started HCV treatment after a positive HCV RNA pre-treatment test. Patients who initiated HCV treatment were followed from their last positive HCV RNA pre-treatment test till their HCV treatment start date and patients who

did not initiate HCV treatment were followed till the censor date of 31st December 2020. HCV treatment response was defined as achieving SVR. Patients who had completed HCV treatment were followed until documentation of SVR or the censoring date of 31st December 2020 for those who did not achieve SVR.

Factors associated with each outcome were analyzed using Fine and Gray's competing risk regression model adjusted for country income level. Death and lost-to follow-up (LTFU) were included as competing risks. LTFU was defined as not seen in clinic for >12 months from the censored date without evidence of transfer. Time-fixed covariates included were age, sex, gender, mode of HIV exposure, and hepatitis B virus (HBV) infection status. Chronic HBV was defined as ever testing positive for HBV surface antigen (HBVsAg). Time-varying covariates included HIV viral load, CD4 cell count, alanine transaminase (ALT), aspartate aminotransferase (AST), creatinine levels above its upper limit of normal, platelet count, and calendar year group of follow-up.

Covariates in univariate analyses with p <0.10 were fitted into multivariate models. The backward stepwise selection process was used and covariates with p <0.05 were considered statistically significant in the multivariate model. All data management and statistical analyses were performed using SAS software version 9.4 (SAS Institute Inc., Cary, NC, USA) and Stata software version 16.1 (StataCorp, College Station, TX, USA).

## Ethical considerations

The following Institutional Review Boards approved the study: Cambodia Ministry of Health National Ethics Committee for Health Research; Kowloon Central / Kowloon East Research Ethics Committee; VHS Institutional Ethics Committee; Institutional Ethics Committee Rao Nursing Home; Kerti Praja Foundation IRB; Severance Hospital Yonsei University College of Medicine IRB; The Ethical Committee for Research in Human Subject, Chiangrai Prachanukroh Hospital; Institutional Review Board Faculty of Medicine, Chulalongkorn University; Committee on Human Rights Related to Research Involving Human Subjects, Faculty of Medicine Ramathibodi Hospital; Hanoi School of Public Health IRB; The Ethical Review Board for Biomedical Research of National Hospital of Tropical Diseases; UNSW Human Research Ethics Committee; and Advarra IRB. Written informed consent was obtained at the following study sites: Chiangrai Prachanukroh Hospital; HIV-NAT, the Thai Red Cross AIDS Research Centre; and Ramathibodi Hospital. Consent at all other sites was waived due to the observational nature of the study, and because data was only collected retrospectively from available medical records.

## Results

### Patient characteristics

A total of 24,421 participants met analysis inclusion criteria. The majority were male (63%) and from lower-middle income countries (74%) (Table 1). Heterosexual contact was the primary reported mode of HIV exposure (78%). Median age at ART initiation was 35 years (Interquartile range [IQR] 29–42). At ART initiation, HIV viral load and CD4 cell count were not tested in 84% and 38%, respectively. Among those with tests available, median HIV viral load at ART initiation was 90,742 copies/mL (IQR 19,118–320,000), and median CD4 cell count was 173 cells/μL (IQR 65–299). Half (49%) were tested for HBVsAg of whom 9% tested positive. Three percent had an ALT test, of whom 8% had levels ≥ 2 times the upper limit of normal (ULN). Two percent had an AST test, of whom almost all (99.8%) had levels ≥ 2 times the ULN. Three percent had a platelet test, of whom 14% had a count <150 x10$^9$/L.

**Table 1. Patient characteristics of total patients included and those with a positive anti-HCV test from 1st January 2010.**

| | Total patients | Total patients with positive anti-HCV (%) |
|---|---|---|
| | **N = 24,421 (100)** | **N = 971 (100)** |
| **Age at ART initiation (years)** | | |
| Median (IQR) | 35 (29–42) | 34 (30–40) |
| ≤30 | 7,214 (30) | 283 (29) |
| 31–40 | 9,982 (41) | 470 (48) |
| 41–50 | 5,060 (21) | 164 (17) |
| >50 | 2,165 (9) | 54 (6) |
| **Sex** | | |
| Male | 15,304 (63) | 789 (81) |
| Female | 9,117 (37) | 182 (19) |
| **Mode of HIV Exposure** | | |
| Heterosexual contact | 19,023 (78) | 459 (47) |
| MSM | 1,895 (8) | 85 (9) |
| Injecting drug use | 754 (3) | 378 (39) |
| Other/unknown | 2,749 (11) | 49 (5) |
| **Viral load at ART initiation (copies/mL)** | | |
| Median (IQR) | 90742 (19118–320000) | 180000 (37600–470000) |
| <1000 | 378 (2) | 15 (2) |
| ≥1000 | 3,537 (14) | 242 (25) |
| Not tested | 20,506 (84) | 714 (74) |
| **CD4 at ART initiation (cells/µL)** | | |
| Median (IQR) | 173 (65–299) | 134 (34–299) |
| ≤200 | 8,594 (35) | 310 (32) |
| 201–350 | 3,941 (16) | 110 (11) |
| 351–500 | 1,499 (6) | 57 (6) |
| >500 | 1,190 (5) | 33 (3) |
| Not tested | 9,197 (38) | 461 (48) |
| **Hepatitis B co-infection** | | |
| Negative | 10,879 (45) | 800 (82) |
| Positive | 1,027 (4) | 114 (12) |
| Not tested | 12,515 (51) | 57 (6) |
| **ALT** | | |
| <2xULN | 620 (3) | 177 (18) |
| 2-5xULN | 43 (0) | 21 (2) |
| >5xULN | 11 (0) | 4 (0) |
| Not tested | 23,747 (97) | 769 (79) |
| **AST** | | |
| <2xULN | 1 (0) | 0 (0) |
| 2-5xULN | 0 (0) | 0 (0) |
| >5xULN | 516 (2) | 146 (15) |
| Not tested | 23,904 (98) | 825 (85) |
| **Platelets ($10^9$/L)** | | |
| <150 | 92 (0) | 28 (3) |
| 150–400 | 534 (2) | 130 (13) |
| >400 | 26 (0) | 14 (1) |
| Not tested | 23769 (97) | 799 (82) |

*(Continued)*

**Table 1.** (Continued)

|  | **Total patients** | **Total patients with positive anti-HCV (%)** |
|---|---|---|
|  | **N = 24,421 (100)** | **N = 971 (100)** |
| **World Bank Country Income Level** |  |  |
| Lower Middle | 17963 (74) | 689 (71) |
| Upper Middle | 4498 (18) | 193 (20) |
| High | 1960 (8) | 89 (9) |

N., number; HCV, hepatitis C virus; ART, Antiretroviral Therapy; IQR, interquartile range; MSM, Men who have sex with men; ALT, alanine aminotransferase; AST, aspartate aminotransferase; ULN, upper limit of normal

Of the total study population, 9169 (38%) had an anti-HCV test from 1st January 2010 onwards, and of these 971 participants (11%) had a positive anti-HCV test result. Of those with a positive anti-HCV test result, the majority were male (81%) and from lower-middle income countries (71%) (Table 1). Although heterosexual contact was the primary mode of HIV exposure (47%), 39% were exposed to HIV through injecting drug use. Their median age at ART initiation was 34 years (IQR 30–40). Among those with tests available, median HIV RNA viral load at ART initiation was higher than the total study population at 180,000 copies/mL (IQR 37,600–470,000), and median CD4 cell count at ART initiation was lower at 134 cells/μL (IQR 34–299). Most (94%) were tested for HBVsAg of whom 12% tested positive. Twenty one percent had an ALT test, of whom 12% had levels $\geq$ 2 times the upper limit of normal (ULN). Fifteen percent had an AST test, of whom all had levels $\geq$ 2 times the ULN. Eighteen percent had a platelet test, of whom 16% had a count $<150$ x$10^9$/L.

## Proportion anti-HCV positive and their HCV cascade of care

Among the 6176 participants with an anti-HCV test in the period 2010 to 2014, 745 (12.1%) had a positive test result. Among the 3090 participants with an anti-HCV test in the period 2015 to 2017, 122 (3.9%) had a positive test, and among the 2715 participants with an anti-HCV test in the period 2018 to 2020, 104 (3.8%) had a positive test (Fig 1).

In the period 2010 to 2014, of the 745 participants with a positive anti-HCV test, 34% had a subsequent HCV RNA and/or HCVcAg test (Fig 2A) and 75% of these had positive results. Of those with a positive HCV RNA and/or HCVcAg test, 66% initiated HCV treatment. Of those who initiated HCV treatment, 98% completed their regimens. Among those who completed HCV treatment, 84% had a subsequent HCV RNA test, of whom 83% achieved SVR. Cascade proportions for the 122 participants and the 104 participants with a positive anti-HCV test in the 2015–2017 (Fig 2B) and the 2018–2020 (Fig 2C) period, were 69% and 80% with a subsequent HCV RNA or HCVcAg test, 87% and 80% with a positive HCV RNA or HCVcAg test result, 59% and 61% initiating HCV treatment, 98% and 90% completing treatment, 95% and 75% with a subsequent HCV RNA test, and 88% and 96% achieving SVR.

## Factors associated with HCV cascade of care outcomes

Factors associated with uptake of screening for chronic HCV are presented in Table 2. HCV screening was less likely among those with older age (31–40 years: HR 0.91, 95% CI 0.83–0.99, p<0.023; 41–50 years: HR 0.72, 95% CI 0.64–0.79, p<0.001; $\geq$50 years: HR 0.63, 95% CI 0.54–0.73, p<0.001 –all vs. 30 years and younger), and those with higher HIV viral loads ($\geq$1000 copies/mL vs. <1000 copies/mL: HR 0.30, 95% CI 0.24–0.37, p<0.001). HCV screening was more likely in later calendar years (2018–2020 vs. 2010–2014: HR 2.23, 95% CI 2.02–2.46,

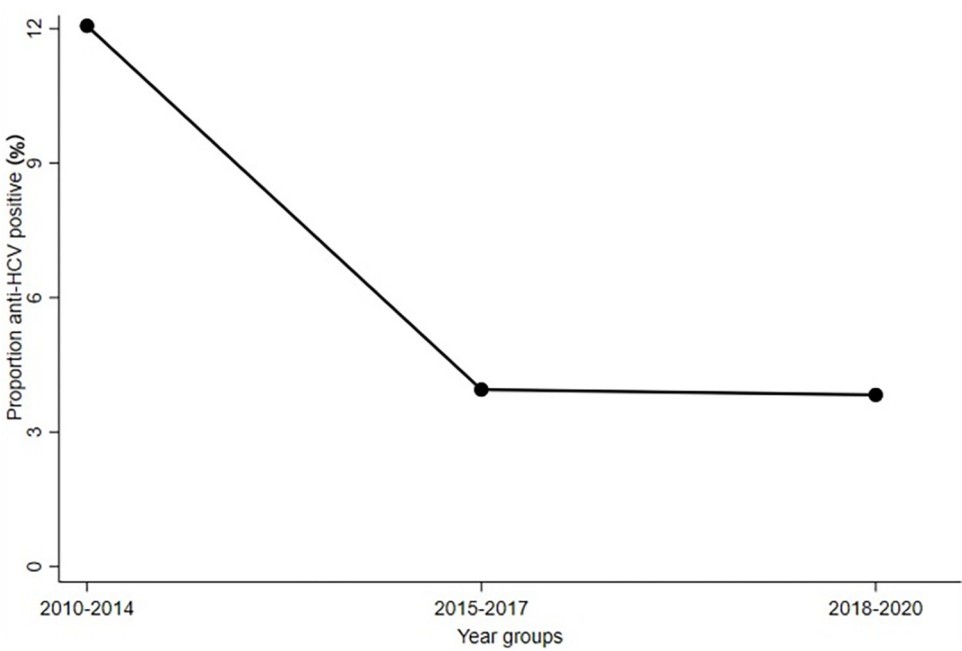

**Fig 1. Proportion anti-HCV positive among those screened for HCV in year groups from 2010 to 2020.**

p<0.001), and in females (vs. males: HR 1.25, 95% CI 1.16–1.35, p<0.001). HCV screening was more likely in those with male-to-male sex (HR 1.52, 95% CI 1.31–1.77, p<0.001) and injecting drug use exposure to HIV (HR 2.65, 95% CI 2.20–3.19, p<0.001) compared to

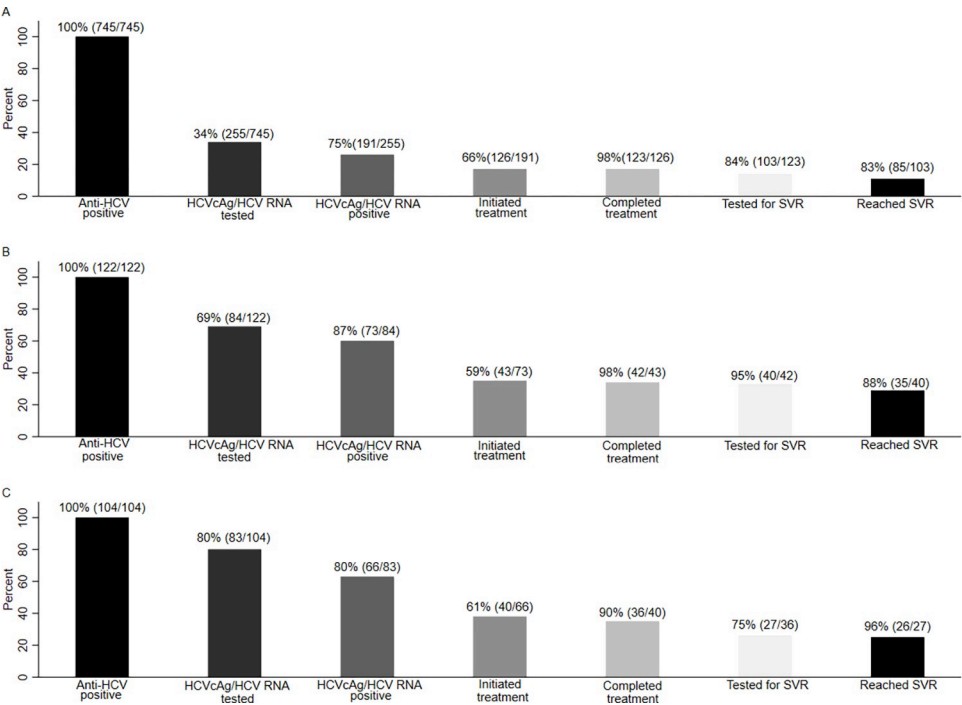

**Fig 2. Hepatitis C cascade for those with positive anti-HCV in year groups from 2010 to 2020.** (A) 2010 to 2014 (B) 2015 to 2017 (C) 2018 to 2020.

**Table 2. Factors associated with uptake of screening for chronic HCV.**

| | No. patients | Follow up (years) | Screened for chronic HCV and with follow up | Incidence rate (per 100 person years) | Univariate | | | Multivariate | | |
|---|---|---|---|---|---|---|---|---|---|---|
| | | | | | HR | 95% CI | p | HR | 95% CI | p |
| **Total** | 15,936 | 76610.75 | 3496 | 4.56 | | | | | | |
| *Calendar year | | | | | | | **<0.001** | | | **<0.001** |
| 2010–2014 | ~ | 38729.26 | 1823 | 4.71 | 1 | | | 1 | | |
| 2015–2017 | ~ | 23868.09 | 482 | 2.02 | 0.89 | (0.80, 0.99) | 0.028 | 1.01 | (0.91, 1.13) | 0.818 |
| 2018–2020 | ~ | 14013.4 | 1191 | 8.5 | 2.69 | (2.48, 2.92) | <0.001 | 2.23 | (2.02, 2.46) | <0.001 |
| **Age at ART initiation (years)** | | | | | | | **<0.001** | | | **<0.001** |
| ≤30 | 4438 | 20826.42 | 1099 | 5.28 | 1 | | | 1 | | |
| 31–40 | 6909 | 34121.51 | 1473 | 4.32 | 0.84 | (0.77, 0.91) | <0.001 | 0.91 | (0.83, 0.99) | 0.023 |
| 41–50 | 3279 | 15663.92 | 668 | 4.26 | 0.79 | (0.72, 0.87) | <0.001 | 0.72 | (0.64, 0.79) | <0.001 |
| 51+ | 1310 | 5998.9 | 256 | 4.27 | 0.76 | (0.67, 0.87) | <0.001 | 0.63 | (0.54, 0.73) | <0.001 |
| **Sex** | | | | | | | | | | |
| Male | 9742 | 46548.3 | 1995 | 4.29 | 1 | | | 1 | | |
| Female | 6194 | 30062.44 | 1501 | 4.99 | 1.2 | (1.12, 1.28) | <0.001 | 1.25 | (1.16, 1.35) | **<0.001** |
| **Mode of HIV Exposure** | | | | | | | **<0.001** | | | **<0.001** |
| Heterosexual contact | 12902 | 61946.26 | 2738 | 4.42 | 1 | | | 1 | | |
| MSM | 557 | 1975.36 | 306 | 15.49 | 3.46 | (3.06, 3.90) | <0.001 | 1.52 | (1.31, 1.77) | <0.001 |
| Injecting drug use | 493 | 2116.04 | 236 | 11.15 | 3.2 | (2.75, 3.72) | <0.001 | 2.65 | (2.20, 3.19) | <0.001 |
| Other/unknown | 1984 | 10573.08 | 216 | 2.04 | 0.49 | (0.43, 0.56) | <0.001 | 0.61 | (0.54, 0.71) | <0.001 |
| *Viral Load (copies/mL) | | | | | | | | | | |
| <1000 | ~ | 33524.61 | 1734 | 5.17 | 1 | | | 1 | | |
| ≥1000 | ~ | 4696.11 | 91 | 1.94 | 0.2 | (0.36, 0.41) | <0.001 | 0.3 | (0.24, 0.37) | **<0.001** |
| Not tested | ~ | 38390.03 | 1671 | 4.35 | | | | | | |
| *CD4 (cells/µL) | | | | | | | <0.001 | | | 0.006 |
| ≤200 | ~ | 8430.88 | 309 | 3.67 | 0.78 | (0.67, 0.91) | 0.002 | 1.17 | (1.00, 1.37) | 0.056 |
| 201–350 | ~ | 11952.75 | 344 | 2.88 | 1 | | | 1 | | |
| 351–500 | ~ | 14156.26 | 527 | 3.72 | 1.64 | (1.43, 1.88) | <0.001 | 1.22 | (1.06, 1.41) | 0.004 |
| >500 | ~ | 29673.4 | 988 | 3.33 | 1.69 | (1.49, 1.91) | <0.001 | 1.04 | (0.92, 1.18) | 0.55 |
| Not tested | ~ | 12397.45 | 1328 | 10.71 | | | | | | |
| **Hepatitis B co-infection** | | | | | | | | | | |
| Negative | 3147 | 9609.33 | 2145 | 22.32 | 1 | | | | | |
| Positive | 394 | 1200.53 | 282 | 23.49 | 1.11 | (0.97, 1.27) | 0.139 | | | |
| Not tested | 12395 | 65800.89 | 1069 | 1.62 | | | | | | |

*(Continued)*

**Table 2.** (Continued)

| | No. patients | Follow up (years) | Screened for chronic HCV and with follow up | Incidence rate (per 100 person years) | Univariate | | | Multivariate | | |
|---|---|---|---|---|---|---|---|---|---|---|
| | | | | | HR | 95% CI | p | HR | 95% CI | p |
| *ALT | | | | | | | <0.001 | | | **<0.001** |
| <2xULN | ~ | 3034.22 | 378 | 12.46 | 1 | | | 1 | | |
| 2-5xULN | ~ | 249.34 | 30 | 12.03 | 1.25 | (0.84, 1.85) | 0.271 | 1.74 | (1.15, 2.62) | 0.008 |
| >5xULN | ~ | 21.25 | 13 | 61.17 | 4.33 | (2.20, 8.49) | <0.001 | 7.1 | (3.64, 13.85) | <0.001 |
| Not tested | ~ | 73305.94 | 3075 | 4.19 | | | | | | |
| *AST | | | | | | | 0.474 | | | |
| <2xULN | ~ | 1954.5 | 287 | 14.68 | 1 | | | | | |
| 2-5xULN | ~ | 221.68 | 25 | 11.28 | 0.76 | (0.49, 1.18) | 0.222 | | | |
| >5xULN | ~ | 11.05 | 4 | 36.19 | 0.99 | (0.28, 3.56) | 0.988 | | | |
| Not tested | ~ | 74423.52 | 3180 | 4.27 | | | | | | |
| *Platelets (10$^9$/L) | | | | | | | 0.039 | | | **0.017** |
| <150 | ~ | 307.62 | 18 | 5.85 | 1 | | | 1 | | |
| 150–400 | ~ | 2951.67 | 378 | 12.81 | 1.83 | (1.13, 2.96) | 0.014 | 2.12 | (1.25, 3.61) | 0.005 |
| >400 | ~ | 96.19 | 17 | 17.67 | 2.09 | (1.08, 4.05) | 0.029 | 2.68 | (1.36, 5.29) | 0.004 |
| Not tested | ~ | 73255.27 | 3083 | 4.21 | | | | | | |
| World Bank Country Income | | | | | | | <0.001 | | | **<0.001** |
| Lower Middle | 12830 | 61189.68 | 1853 | 3.03 | 1 | | | 1 | | |
| Upper Middle | 2547 | 12955.58 | 1297 | 10.01 | 4.68 | (4.36, 5.04) | <0.001 | 2.68 | (2.45, 2.93) | <0.001 |
| High | 559 | 2465.49 | 346 | 14.03 | 5.92 | (5.30, 6.61) | <0.001 | 3.45 | (2.95, 4.03) | <0.001 |

N., number; HCV, hepatitis C virus; ART, Antiretroviral Therapy; IQR, interquartile range; MSM, Men who have sex with men; ALT, alanine aminotransferase; AST, aspartate aminotransferase; ULN, upper limit of normal

*time-updated variables

Not tested values were included in the analysis as a separate category but were excluded from test for heterogeneity

heterosexual HIV exposure, and in those with higher CD4 cell counts (351–500 cells/μL vs. 200–350 cells/μL: HR 1.22, 95% CI 1.06–1.41, p = 0.004). Those with higher ALT levels were more likely to have HCV screening (vs. <2 times ULN; ALT 2–5 times ULN: HR 1.74, 95% CI 1.15–2.62, p = 0.008; ALT >5 times ULN: HR 7.10, 95% CI 3.64–13.85, p<0.001), as were those with higher platelet counts (vs. <150 x10$^9$/L; 150–400: HR 2.12, 95% CI 1.25–3.61, p = 0.005; ≥400: HR 2.68, 95% CI 1.36–5.29, p = 0.004). Study participants from sites in high income (HR 3.45, 95% CI 2.95–4.03, p<0.001) and upper-middle income countries (HR 2.68, 95% CI 2.45–2.93, p<0.001) were more likely to have HCV screening than those in lower-middle income countries.

Chronic HCV treatment initiation was less likely among those with male-to-male sex (HR 0.50, 95% CI 0.28–0.89, p = 0.018) and injecting drug use HIV exposure (HR 0.52, 95% CI 0.30–0.91, p = 0.021) compared to those with heterosexual HIV exposure, in those with lower CD4 cell counts (≤200 cells/μL vs. 201–350 cells/μL; HR 0.34, 95% CI 0.14–0.79, p = 0.013),

and those from study sites in upper-middle income countries (vs. lower-middle income countries; HR 0.47, 95% CI 0.31–0.71, p<0.001) (Table 3). HCV treatment initiation was more likely in later calendar years (vs. 2010–2015; 2015–2017: HR 5.18, 95% CI 2.49–10.76, p<0.001; 2018–2020: HR 5.74, 95% CI 2.76–11.93, p<0.001).

Factors associated with SVR are presented in Table 4. SVR was more likely in later calendar years (vs. 2010–2014; 2015–2017: HR 9.84; 95% CI 1.36–70.95, p = 0.023; 2018–2020: HR 32.53; 95% CI 4.74–253.43; p<0.001), and in those from upper-middle income (HR 2.51; 95% CI 1.62–3.90, p<0.001) and high income country sites (HR 1.84; 95% CI 1.14–2.95; p = 0.012), compared to lower-middle income country sites.

### Probability of not initiating HCV treatment after positive anti-HCV

The probability of not initiating HCV treatment at one year after a positive anti-HCV, was 0.98 for those screening positive in 2010–2014, 0.90 for those screening positive in 2015–2017, and 0.67 for those screening positive in 2018–2020 (Fig 3). This represented an 8% decrease in the probability of not initiating treatment between 2010–2014 to 2015–2017, and a 23% decrease from 2015–2017 to 2018–2020.

## Discussion

In our analysis of HCV coinfection among adult PLHIV across 7 countries in the Asia-Pacific region from 2010 to 2020, we found low chronic HCV screening rates, and a decrease in anti-HCV prevalence. By the DAA era, certain elements of the HCV cascade of care improved, such as pre-HCV treatment testing and achieving SVR, but treatment initiation worsened. We found having HCV in later calendar years and at HIC sites were associated with increased chronic HCV screening, treatment initiation and/or achieving SVR. Older age, HIV exposure through male-to-male sex or injecting drug use, lower CD4 cell count, and higher HIV viral load were associated with reduced HCV screening or treatment.

The 12.1% prevalence of anti-HCV positives in our cohort between 2010 and 2014 is consistent with findings from other regional analyses in that time period: 11.9% of Australian PLHIV tested, and 15.2% of those tested in a regional adult HIV cohort were anti-HCV positive [27, 34]. Declines in the prevalence of anti-HCV similar those we observed from 2010 to 2020, have been well documented among other Asian PLHIV cohorts, and are likely due to changes in age and HIV risk group composition and improved access to interventions that reduce HCV transmission over time [11, 14].

The 62% of our adult PLHIV cohort without an anti-HCV test result available from January 2010, is consistent with poor chronic HCV screening rates in other adult PLHIV cohorts in the region [22, 23], and in contrast to higher HCV screening rates in better resourced countries in the region [16, 34].

Whilst our findings of substantial improvements in pre-treatment HCV RNA testing and the proportions achieving SVR in the DAA versus pre-DAA era are consistent with improvements documented in HIC and Western PLHIV cohorts, these cohorts also documented substantial improvement in the HIV/HCV co-infected proportions initiating HCV treatment, which we did not. For example, a Swiss HIV cohort analysis found a continuous increase in HCV treatment uptake from 2009 to 2015, a time period corresponding to pre-DAA and then to 2nd generation DAA availability in the country [35]. A recent analysis among PLHIV in Australia documented an increase in HCV RNA testing uptake from 85% in 2010–2015 (their pre-DAA era) to 93% in 2016–2018, and an increase from 7% to 73% in those eligible for HCV treatment receiving it [36].

**Table 3. Factors associated with chronic HCV treatment initiation.**

| | No. patients | Follow up (years) | Initiated chronic HCV treatment | Incidence rate (per 100 person years) | Univariate | | | Multivariate | | |
|---|---|---|---|---|---|---|---|---|---|---|
| | | | | | HR | 95% CI | p | HR | 95% CI | p |
| **Total** | 269 | 568.6 | 179 | 31.48 | | | | | | |
| *Calendar year | | | | | | | <0.001 | | | **<0.001** |
| 2010–2014 | ~ | 124.99 | 9 | 7.2 | 1 | | | 1 | | |
| 2015–2017 | ~ | 218.87 | 67 | 30.61 | 7.51 | (3.73, 15.14) | <0.001 | 5.18 | (2.49, 10.76) | <0.001 |
| 2018–2020 | ~ | 224.74 | 103 | 45.83 | 8.77 | (4.40, 17.50) | <0.001 | 5.74 | (2.76, 11.93) | <0.001 |
| **Age at ART initiation (years)** | | | | | | | 0.152 | | | |
| ≤30 | 68 | 131.71 | 40 | 30.37 | 1 | | | | | |
| 31–40 | 132 | 329.44 | 86 | 26.11 | 0.94 | (0.64, 1.39) | 0.769 | | | |
| 41–50 | 53 | 87.89 | 41 | 46.65 | 1.36 | (0.89, 2.09) | 0.156 | | | |
| 51+ | 16 | 19.57 | 12 | 61.33 | 1.42 | (0.70, 2.90) | 0.335 | | | |
| **Sex** | | | | | | | | | | |
| Male | 209 | 440.45 | 131 | 29.74 | 1 | | | | | |
| Female | 60 | 128.15 | 48 | 37.46 | 1.35 | (0.99, 1.85) | 0.06 | | | |
| **Mode of HIV Exposure** | | | | | | | 0.069 | | | 0.015 |
| Heterosexual contact | 154 | 331.7 | 112 | 33.77 | 1 | | | 1 | | |
| MSM | 51 | 73.93 | 28 | 37.87 | 0.84 | (0.55, 1.26) | 0.392 | 0.5 | (0.28, 0.89) | 0.018 |
| Injecting drug use | 48 | 147.57 | 27 | 18.3 | 0.64 | (0.42, 0.99) | 0.043 | 0.52 | (0.30, 0.91) | 0.021 |
| Other/unknown | 16 | 15.4 | 12 | 77.93 | 1.43 | (0.83, 2.46) | 0.2 | 0.87 | (0.47, 1.62) | 0.659 |
| *Viral Load (copies/mL) | | | | | | | | | | |
| <1000 | ~ | 381.13 | 134 | 35.16 | 1 | | | | | |
| ≥1000 | ~ | 18.51 | 4 | 21.61 | 0.32 | (0.11, 0.88) | 0.028 | | | |
| Not tested | ~ | 168.97 | 41 | 24.27 | | | | | | |
| *CD4 (cells/µL) | | | | | | | <0.001 | | | **0.002** |
| ≤200 | ~ | 100.31 | 6 | 5.98 | 0.24 | (0.10, 0.57) | 0.001 | 0.34 | (0.14, 0.79) | 0.013 |
| 201–350 | ~ | 117.35 | 24 | 20.45 | 1 | | | 1 | | |
| 351–500 | ~ | 107.41 | 38 | 35.38 | 1.73 | (1.00, 3.00) | 0.05 | 1.56 | (0.87, 2.77) | 0.134 |
| >500 | ~ | 158 | 52 | 32.91 | 1.93 | (1.19, 3.12) | 0.007 | 1.57 | (0.95, 2.59) | 0.075 |
| Not tested | ~ | 85.52 | 59 | 68.99 | | | | | | |
| **Hepatitis B co-infection** | | | | | | | | | | |
| Negative | 217 | 486.36 | 141 | 28.99 | 1 | | | | | |
| Positive | 21 | 45.83 | 16 | 34.91 | 1.29 | (0.79, 2.10) | 0.304 | | | |
| Not tested | 31 | 36.42 | 22 | 60.41 | | | | | | |
| *ALT | | | | | | | 0.191 | | | |

(Continued)

**Table 3.** (Continued)

| | No. patients | Follow up (years) | Initiated chronic HCV treatment | Incidence rate (per 100 person years) | Univariate | | | Multivariate | | |
|---|---|---|---|---|---|---|---|---|---|---|
| | | | | | HR | 95% CI | p | HR | 95% CI | p |
| <2xULN | ~ | 301.37 | 79 | 26.21 | 1 | | | | | |
| 2-5xULN | ~ | 60.6 | 33 | 54.46 | 1.45 | (0.92, 2.28) | 0.112 | | | |
| >5xULN | ~ | 7.62 | 6 | 78.75 | 1.58 | (0.71, 3.49) | 0.259 | | | |
| Not tested | ~ | 199.02 | 61 | 30.65 | | | | | | |
| *AST | | | | | | | 0.721 | | | |
| <2xULN | ~ | 241.03 | 68 | 28.21 | 1 | | | | | |
| 2-5xULN | ~ | 52.38 | 27 | 51.54 | 1.21 | (0.75, 1.96) | 0.428 | | | |
| >5xULN | ~ | 9.76 | 4 | 40.97 | 0.95 | (0.32, 2.80) | 0.922 | | | |
| Not tested | ~ | 265.42 | 80 | 30.14 | | | | | | |
| *Platelets ($10^9$/L) | | | | | | | 0.483 | | | |
| <150 | ~ | 60.57 | 17 | 28.06 | 1 | | | | | |
| 150–400 | ~ | 291.04 | 100 | 34.36 | 1.12 | (0.63, 2.00) | 0.694 | | | |
| >400 | ~ | 4.26 | 1 | 23.49 | 0.42 | (0.07, 2.40) | 0.326 | | | |
| Not tested | ~ | 212.73 | 61 | 28.68 | | | | | | |
| **World Bank Country Income** | | | | | | | <0.001 | | | **<0.001** |
| Lower Middle | 114 | 247.87 | 85 | 34.29 | 1 | | | 1 | | |
| Upper Middle | 95 | 263.5 | 55 | 20.87 | 0.55 | (0.40, 0.76) | <0.001 | 0.47 | (0.31, 0.71) | <0.001 |
| High | 60 | 57.24 | 39 | 68.13 | 1.03 | (0.69, 1.53) | 0.897 | 1.14 | (0.61, 2.11) | 0.685 |

N., number; HCV, hepatitis C virus; ART, Antiretroviral Therapy; IQR, interquartile range; MSM, Men who have sex with men; ALT, alanine aminotransferase; AST, aspartate aminotransferase; ULN, upper limit of normal

*time-updated variables

Not tested values were included in the analysis as a separate category but were excluded from test for heterogeneity

The current SVR rate of 96% achieved in our cohort is comparable to the 86–98% range documented in high income countries [37]. However the current proportions of those with pre-treatment HCV RNA testing (80%) and those initiating HCV treatment (61%) in our cohort fall well below those documented in HIC cohorts: 93% of those anti-HCV positive in a European HIV cohort had a subsequent HCV RNA test [38], and 73% of Australian HCV co-infected PLHIV initiated treatment [36]. It should be noted that HCV cascade performance in the later year of our analysis might have been affected by local COVID-19 epidemics, which negatively impacted chronic HCV screening, treatment initiation, and hepatitis elimination programs in other settings [39, 40].

Whilst we found older age and having a higher HIV viral load were associated only with reduced HCV screening, these have been associated with other negative HCV cascade outcomes. Older age was associated with not receiving HCV RNA testing [36], and reduced odds of receiving HCV DAA treatment [41]. Having a detectable HIV viral load was a predictor for not being referred for HCV therapy following a positive screening in an analysis of adult

**Table 4. Factors associated with chronic HCV treatment response.**

| | No. patients | Follow up (years) | Reached SVR | Incidence rate (per 100 person years) | Univariate | | | Multivariate | | |
|---|---|---|---|---|---|---|---|---|---|---|
| | | | | | HR | 95% CI | p | HR | 95% CI | p |
| **Total** | 218 | 383.14 | 124 | 32.36 | | | | | | |
| *\*Calendar year* | | | | | | | <0.001 | | | **<0.001** |
| 2010–2014 | ~ | 53.81 | 1 | 1.86 | 1 | | | 1 | | |
| 2015–2017 | ~ | 128.99 | 25 | 19.38 | 11.27 | (1.55, 82.07) | 0.017 | 9.84 | (1.36, 70.95) | 0.023 |
| 2018–2020 | ~ | 200.34 | 98 | 48.92 | 36.26 | (5.19, 253.43) | <0.001 | 32.53 | (4.74, 223.43) | <0.001 |
| **Age at ART initiation (years)** | | | | | | | 0.136 | | | |
| ≤30 | 38 | 58.88 | 21 | 35.67 | 1 | | | | | |
| 31–40 | 112 | 217.63 | 58 | 26.65 | 0.83 | (0.51, 1.35) | 0.449 | | | |
| 41–50 | 53 | 92.1 | 35 | 38 | 1.4 | (0.80, 2.43) | 0.235 | | | |
| 51+ | 15 | 14.52 | 10 | 68.88 | 1.06 | (0.55, 2.04) | 0.864 | | | |
| **Sex** | | | | | | | | | | |
| Male | 157 | 277.94 | 90 | 32.38 | 1 | | | | | |
| Female | 61 | 105.2 | 34 | 32.32 | 0.89 | (0.60, 1.31) | 0.54 | | | |
| **Mode of HIV Exposure** | | | | | | | 0.553 | | | |
| Heterosexual contact | 132 | 228.24 | 81 | 35.49 | 1 | | | | | |
| MSM | 27 | 31.09 | 17 | 54.69 | 1.09 | (0.70, 1.70) | 0.697 | | | |
| Injecting drug use | 38 | 79.12 | 16 | 20.22 | 0.71 | (0.39, 1.30) | 0.268 | | | |
| Other/unknown | 21 | 44.7 | 10 | 22.37 | 0.77 | (0.39, 1.50) | 0.442 | | | |
| *\*Viral Load (copies/mL)* | | | | | | | | | | |
| <1000 | ~ | 217.45 | 85 | 39.09 | 1 | | | | | |
| ≥1000 | ~ | 12.71 | 2 | 15.73 | 0.28 | (0.06, 1.25) | 0.096 | | | |
| Not tested | ~ | 152.98 | 37 | 24.19 | | | | | | |
| *\*CD4 (cells/μL)* | | | | | | | 0.021 | | | |
| ≤200 | ~ | 33 | 3 | 9.09 | 0.23 | (0.07, 0.82) | 0.024 | | | |
| 201–350 | ~ | 46 | 15 | 32.61 | 1 | | | | | |
| 351–500 | ~ | 76.54 | 23 | 30.05 | 1.44 | (0.71, 2.91) | 0.313 | | | |
| >500 | ~ | 118.62 | 29 | 24.45 | 1.5 | (0.78, 2.89) | 0.222 | | | |
| Not tested | ~ | 108.97 | 54 | 49.55 | | | | | | |
| **Hepatitis B co-infection** | | | | | | | | | | |
| Negative | 173 | 321.27 | 94 | 29.26 | 1 | | | | | |
| Positive | 18 | 19.47 | 12 | 61.62 | 1.57 | (0.84, 2.92) | 0.159 | | | |
| Not tested | 27 | 42.39 | 18 | 42.46 | | | | | | |
| *\*ALT* | | | | | | | | | | |
| <2xULN | ~ | 163.18 | 92 | 56.38 | 1 | | | | | |
| ≥2 ULN | ~ | 14.85 | 5 | 35.84 | 0.67 | (0.28, 1.61) | 0.369 | | | |
| Not tested | ~ | 205.11 | 27 | 13.16 | | | | | | |
| *\*AST* | | | | | | | | | | |
| <2xULN | ~ | 181.84 | 85 | 46.74 | 1 | | | | | |
| ≥2 ULN | ~ | 11.04 | 7 | 64.47 | 1.63 | (0.73, 3.65) | 0.238 | | | |
| Not tested | ~ | 190.26 | 32 | 16.82 | | | | | | |
| *\*Platelets* | | | | | | | 0.294 | | | |
| <150 | ~ | 24.02 | 13 | 54.13 | 1 | | | | | |
| 150–400 | ~ | 162.43 | 84 | 51.71 | 1.28 | (0.74, 2.24) | 0.379 | | | |

*(Continued)*

**Table 4.** (Continued)

| | No. patients | Follow up (years) | Reached SVR | Incidence rate (per 100 person years) | Univariate | | | Multivariate | | |
|---|---|---|---|---|---|---|---|---|---|---|
| | | | | | HR | 95% CI | p | HR | 95% CI | p |
| >400 | ~ | 8.81 | 1 | 11.35 | 0.4 | (0.07, 2.36) | 0.309 | | | |
| Not tested | ~ | 187.88 | 26 | 13.84 | | | | | | |
| *Creatinine | | | | | | | | | | |
| <1.0 x ULN | ~ | 195.09 | 89 | 45.62 | 1 | | | | | |
| 1.0–3.0 x ULN | ~ | 16.11 | 8 | 49.67 | 1.11 | (0.52, 2.35) | 0.793 | | | |
| Not tested | ~ | 171.94 | 27 | 15.70 | | | | | | |
| World Bank Country Income | | | | | | | <0.001 | | | **0.001** |
| Lower Middle | 129 | 283.86 | 54 | 19.02 | 1 | | | 1 | | |
| Upper Middle | 55 | 52.15 | 47 | 90.13 | 2.73 | (1.87, 4.00) | <0.001 | 2.51 | (1.62, 3.90) | <0.001 |
| High | 34 | 47.13 | 23 | 48.8 | 1.76 | (1.14, 2.72) | 0.011 | 1.84 | (1.14, 2.95) | 0.012 |

N., number; HCV, hepatitis C virus; ART, Antiretroviral Therapy; IQR, interquartile range; MSM, Men who have sex with men; ALT, alanine aminotransferase; AST, aspartate aminotransferase; ULN, upper limit of normal

*time-updated variables

Not tested values were included in the analysis as a separate category but were excluded from test for heterogeneity

PLHIV in the US [42]. Our finding that HCV screening was more likely in females compared to males contrasts with other studies in which men were more likely to have good cascade outcomes, including receiving HCV RNA testing [36], and treatment initiation [43]. Although poor immune status has been identified as a predictor for failure to achieve SVR following

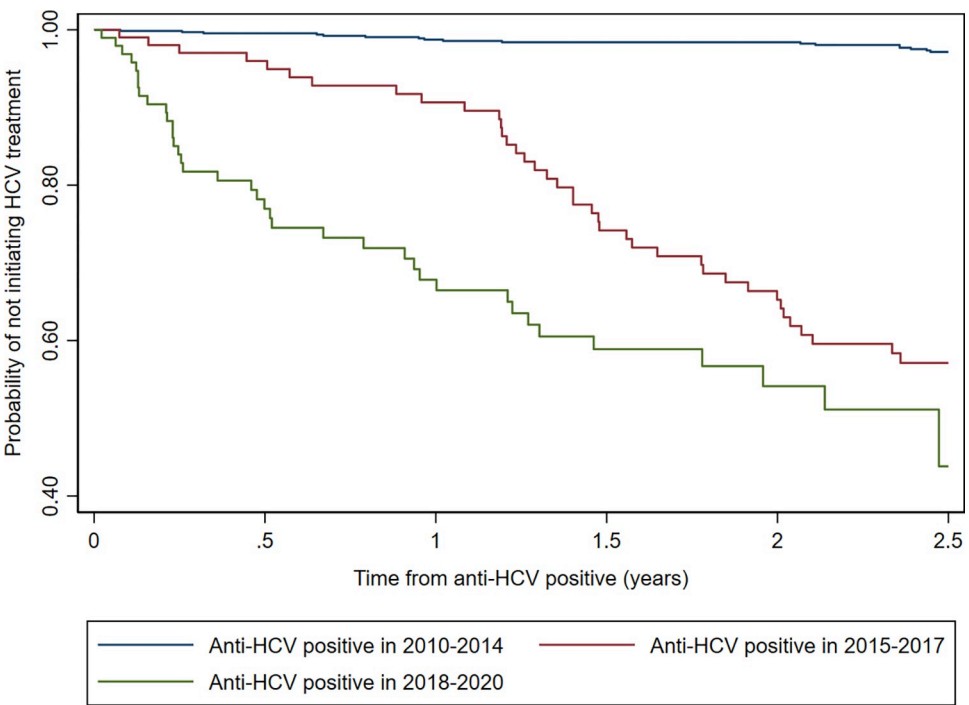

**Fig 3. Probability of not initiating HCV treatment after positive anti-HCV.**

DAA treatment [44], our finding of an association between lower CD4 counts and reduced HCV screening and treatment initiation has not been widely documented.

Our finding that chronic HCV treatment initiation was less likely among those with injecting drug use HIV exposure is consistent with other studies. A recent analysis of the hepatitis C treatment cascade in the US found that those who injected drugs or used other illicit substances in the past 6 months were less likely to initiate HCV treatment [43], and an analysis of adult HIV/HCV co-infected patients in the DAA era in the Netherlands found lower proportions of those who injected drugs initiating HCV treatment [45]. Among a PLHIV cohort in Israel, drug use was also found to be associated with reduced access to HCV treatment [46] and active injection drug use was associated with reduced odds of linkage to HCV care among PLHIV in the US [47]. Whilst the association between injecting drug use and reduced odds of HCV treatment initiation or linkage is likely a reflection of the various socio-economic, health system, and policy factors that contribute to reduced access to health services among PWID, it is important to note the substantial research showing that PWID achieve good HCV cascade outcomes, including adherence to treatment, low discontinuation rates, and high SVR rates [48–50]. Contrary to finding that HCV treatment initiation was less likely among MSM in our region, Boerekamps et al. found HCV treatment uptake to be highest among MSM relative to other populations [45].

In addition, our analysis highlighted gaps in testing capacities with high proportions of our HIV/HCV co-infected cohort having no ALT, AST or platelet test results available. These, and the other persistent gaps in the chronic HCV cascade of care we identified are particularly concerning given the burden of HCV/HIV co-infection in the Asia-Pacific region and emphasize the need for focused efforts to strengthen chronic HCV screening, pre-treatment testing, HCV treatment initiation, and monitoring. In support of these efforts, novel interventions should be considered or scaled up in the region, including point-of-care antibody testing [51], reducing diagnostic and DAA costs [52], non-specialist-led HCV care and treatment [53, 54] and community-led same day test and treat models [55]. Improved HCV screening and scale-up of HCV treatment are also important for reducing the risk of HCV re-infection amongst PLHIV [56]. A recent study of HCV re-infection among an adult PLHIV cohort in Taiwan found that those living with HIV had higher HCV re-infection rates than the HIV-negative reference population, and that HCV reinfection incidence increased following the availability of DAAs in the country, further highlighting the importance of post HCV treatment surveillance among PLHIV [57].

The findings of our analysis should be considered in the context of a number of limitations. We enrolled relatively small numbers of patients in the 2018–2020 time period. In addition, the data collected are limited to those retained in HV care, a subpopulation that might have better health-seeking behaviors and outcomes and may not be representative of the national HIV/HCV co-infected population. Despite these limitations, our analysis included a substantial number of participants from a number of countries in the region and provides an informative picture of HCV coinfection and cascade performance among adult PLHIV in the Asia-Pacific region.

## Acknowledgments

TAHOD-LITE study members: V Khol, V Ouk, C Pov, National Center for HIV/AIDS, Dermatology & STDs, Phnom Penh, Cambodia; MP Lee, PCK Li, TS Kwong, TH Li, Queen Elizabeth Hospital, Hong Kong SAR; N Kumarasamy, C Ezhilarasi, Chennai Antiviral Research and Treatment Clinical Research Site (CART CRS), VHS-Infectious Diseases Medical Centre, VHS, Chennai, India; S Pujari, K Joshi, S Gaikwad, A Chitalikar, Institute of Infectious

Diseases, Pune, India; IKA Somia, TP Merati, AAS Sawitri, F Yuliana, Faculty of Medicine Udayana University—Prof. Dr. I.G.N.G. Ngoerah Hospital, Bali, Indonesia; JY Choi, Na S, JM Kim, Division of Infectious Diseases, Department of Internal Medicine, Yonsei University College of Medicine, Seoul, South Korea; CD Do, AV Ngo, LT Nguyen, Bach Mai Hospital, Hanoi, Vietnam; TN Pham, KV Nguyen, DTH Nguyen, DT Nguyen, National Hospital for Tropical Diseases, Hanoi, Vietnam; A Avihingsanon, S Gatechompol, P Phanuphak, C Pha-dungphon, HIV-NAT/Thai Red Cross AIDS Research Centre, Bangkok, Thailand; S Kiertibur-anakul, A Phuphuakrat, L Chumla, N Sanmeema, Faculty of Medicine Ramathibodi Hospital, Mahidol University, Bangkok, Thailand; S Khusuwan, P Kambua, S Pongprapass, J Limlertch-areonwanit, Chiangrai Prachanukroh Hospital, Chiang Rai, Thailand; AH Sohn, JL Ross*, B Petersen, TREAT Asia, amfAR—The Foundation for AIDS Research, Bangkok, Thailand; MG Law, A Jiamsakul, D Rupasinghe, The Kirby Institute, UNSW Sydney, NSW, Australia.

* denotes lead author for this group (contact email: jeremy.ross@treatasia.org)

## Author Contributions

**Conceptualization:** Jeremy Ross.

**Formal analysis:** Dhanushi Rupasinghe.

**Methodology:** Jeremy Ross, Anchalee Avihingsanon, Man Po Lee, Sanjay Pujari, Nagalingeswaran Kumarasamy, Suwimon Khusuwan, Vohith Khol, I. Ketut Agus Somia, Thach Ngoc Pham, Sasisopin Kiertiburanakul, Jun Yong Choi, Cuong Duy Do.

**Supervision:** Awachana Jiamsakul.

**Writing – original draft:** Jeremy Ross, Dhanushi Rupasinghe.

**Writing – review & editing:** Jeremy Ross, Dhanushi Rupasinghe, Anchalee Avihingsanon, Man Po Lee, Sanjay Pujari, Gerald Sharp, Nagalingeswaran Kumarasamy, Suwimon Khusuwan, Vohith Khol, I. Ketut Agus Somia, Thach Ngoc Pham, Sasisopin Kiertiburanakul, Jun Yong Choi, Cuong Duy Do, Annette H. Sohn.

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
