## [Decision Letter · Decision Letter 0]

27 Mar 2023

PONE-D-23-04685Trends in hepatitis C virus coinfection and its cascade of care among adults living with HIV in Asia between 2010 and 2020PLOS ONE

Dear Dr. Ross,

Thank you for submitting your manuscript to PLOS ONE. After careful consideration, we feel that it has merit but does not fully meet PLOS ONE’s publication criteria as it currently stands. Therefore, we invite you to submit a revised version of the manuscript that addresses the points raised during the review process. Please submit your revised manuscript by May 11 2023 11:59PM. If you will need more time than this to complete your revisions, please reply to this message or contact the journal office at plosone@plos.org. Please include the following items when submitting your revised manuscript:A rebuttal letter that responds to each point raised by the academic editor and reviewer(s). You should upload this letter as a separate file labeled 'Response to Reviewers'.A marked-up copy of your manuscript that highlights changes made to the original version. You should upload this as a separate file labeled 'Revised Manuscript with Track Changes'.An unmarked version of your revised paper without tracked changes. You should upload this as a separate file labeled 'Manuscript'.If applicable, we recommend that you deposit your laboratory protocols in protocols.io to enhance the reproducibility of your results. Protocols.io assigns your protocol its own identifier (DOI) so that it can be cited independently in the future. For instructions see: https://journals.plos.org/plosone/s/submission-guidelines#loc-laboratory-protocols. Additionally, PLOS ONE offers an option for publishing peer-reviewed Lab Protocol articles, which describe protocols hosted on protocols.io. Read more information on sharing protocols at https://plos.org/protocols?utm_medium=editorial-email&utm_source=authorletters&utm_campaign=protocols.

We look forward to receiving your revised manuscript.

Kind regards,

Chen-Hua Liu

Academic Editor

PLOS ONE

Journal Requirements:

   "AHS has received grants to her institution from ViiV Healthcare and Gilead Sciences. Other authors have declared that no competing interests exist. "

4. One of the noted authors is a group or consortium " TAHOD-LITE study group of IeDEA Asia‐Pacific". In addition to naming the author group, please list the individual authors and affiliations within this group in the acknowledgments section of your manuscript. Please also indicate clearly a lead author for this group along with a contact email address.

Reviewers' comments:

Reviewer's Responses to Questions

**Comments to the Author**

1. Is the manuscript technically sound, and do the data support the conclusions?

Reviewer #1: Partly

2. Has the statistical analysis been performed appropriately and rigorously? 

Reviewer #1: Yes

3. Have the authors made all data underlying the findings in their manuscript fully available?

Reviewer #1: Yes

4. Is the manuscript presented in an intelligible fashion and written in standard English?

Reviewer #1: Yes

5. Review Comments to the Author

Reviewer #1: This study from a prospective observational cohort of the International Epidemiology Databases to Evaluate AIDS (IeDEA) Asia-Pacific (TAHOD-LITE) was aimed to assessed regional HCV coinfection and cascade outcomes among adults living with HIV in care from 2010-2020 in Asia.

1. The proportion of people living with HIV (PLHIV) with a positive anti-HCV test was reported over three time periods that broadly correspond to very limited DAA access (the pre-DAA era), initial DAA access, and expanded DAA access in the Asia-Pacific region: 2010-2014, 2015-2017, and 2018-2020. Results of three time periods should be mentioned in Abstract.

2. Liver fibrosis, liver function reserve, renal function, HCV genotype and HCV RNA may be factors associated with chronic HCV treatment response. The proportion of liver cirrhosis, liver fibrosis grade, serum albumin, creatine, HCV genotype and HCV RNA should be analyzed in Table 4.

3. Table 2, Calendar year, “2017”-2020 should be “2018”-2020.

6. PLOS authors have the option to publish the peer review history of their article (what does this mean?). If published, this will include your full peer review and any attached files.

Reviewer #1: No

---

## [Author Response · Author response to Decision Letter 0]

28 Apr 2023

Responses to reviewer and editor comments have been provided in the attached cover letter and response to reviewers letter. These are also included below.

Academic editor comments and responses

Please ensure that your manuscript meets PLOS ONE's style requirements, including those for file naming. Response: We have reviewed and confirm that our manuscript meets PLOS ONE's style requirements, including those for file naming.

Thank you for stating the following in the Competing Interests section: "AHS has received grants to her institution from ViiV Healthcare and Gilead Sciences. Other authors have declared that no competing interests exist.” Please confirm that this does not alter your adherence to all PLOS ONE policies on sharing data and materials, by including the following statement: "This does not alter our adherence to PLOS ONE policies on sharing data and materials.” Response: We have updated our Competing Interests section to include the above statement and provided it in the revised cover letter, as requested. 

We note that you have indicated that data from this study are available upon request. PLOS only allows data to be available upon request if there are legal or ethical restrictions on sharing data publicly. Response: We have provided the ethical or legal restrictions on sharing our de-identified data set in the revised cover letter, as requested.

One of the noted authors is a group or consortium "TAHOD-LITE study group of IeDEA Asia‐Pacific". In addition to naming the author group, please list the individual authors and affiliations within this group in the acknowledgments section of your manuscript. Please also indicate clearly a lead author for this group along with a contact email address. Response: The individual authors for this group and their affiliations have been provided. We indicated a lead author for this group and their contact email address in the revised manuscript, as requested. 

Please include your full ethics statement in the ‘Methods’ section of your manuscript file. In your statement, please include the full name of the IRB or ethics committee who approved or waived your study, as well as whether or not you obtained informed written or verbal consent. If consent was waived for your study, please include this information in your statement as well. Response: We have replaced the previous ethics statement with the following full ethics statement that includes the above information, in the ‘Methods’ section of our manuscript, as requested: “The following IRBs approved the study: Cambodia Ministry of Health National Ethics Committee for Health Research; Kowloon Central / Kowloon East Research Ethics Committee; VHS Institutional Ethics Committee; Institutional Ethics Committee Rao Nursing Home; Kerti Praja Foundation IRB; Severance Hospital Yonsei University College of Medicine IRB; The Ethical Committee for Research in Human Subject, Chiangrai Prachanukroh Hospital; Institutional Review Board Faculty of Medicine, Chulalongkorn University; Committee on Human Rights Related to Research Involving Human Subjects, Faculty of Medicine Ramathibodi Hospital; Hanoi School of Public Health IRB; The Ethical Review Board for Biomedical Research of National Hospital of Tropical Diseases; UNSW Human Research Ethics Committee; and Advarra IRB. Written informed consent was obtained at the following study sites: Chiangrai Prachanukroh Hospital; HIV-NAT, the Thai Red Cross AIDS Research Centre; and Ramathibodi Hospital. Consent at all other sites was waived due to the observational nature of the study, and because data was only collected retrospectively from available medical records.” 

Please review your reference list to ensure that it is complete and correct. Response: We have reviewed the reference list and confirm it is complete and correct.

Reviewer #1 comments and responses

Overall: This study from a prospective observational cohort of the International Epidemiology Databases to Evaluate AIDS (IeDEA) Asia-Pacific (TAHOD-LITE) was aimed to assess regional HCV coinfection and cascade outcomes among adults living with HIV in care from 2010-2020 in Asia.

Response: Thank you for your review of our manuscript and recommendations to improve it. We hope these have been adequately addressed in our responses (below) and associated edits.

Comment 1. The proportion of people living with HIV (PLHIV) with a positive anti-HCV test was reported over three time periods that broadly correspond to very limited DAA access (the pre-DAA era), initial DAA access, and expanded DAA access in the Asia-Pacific region: 2010-2014, 2015-2017, and 2018-2020. Results of three time periods should be mentioned in Abstract.

Response: We have added data on the positive anti-HCV test results and HCV cascade proportions from all three of the 2010-2014, 2015-2017, and 2018-2020 time periods to the abstract. 

Comment 2. Liver fibrosis, liver function reserve, renal function, HCV genotype and HCV RNA may be factors associated with chronic HCV treatment response. The proportion of liver cirrhosis, liver fibrosis grade, serum albumin, creatinine, HCV genotype and HCV RNA should be analyzed in Table 4.

Response: Thank you for this suggestion to include these additional factors in the chronic HCV treatment response regression analysis. As suggested we did have adequate creatinine data and have added that to the analysis. However, we were either unable to include the other factors or felt it could confuse the findings. We believe that associations between liver cirrhosis and fibrosis and chronic HCV treatment response are likely already covered to an extent by our inclusion of ALT and AST levels. Unfortunately, we did not collect albumin test results from our study participants so cannot add that to the regression analysis. In addition, HCV genotype results are very limited or often not available, particularly from sites in resource-limited countries or countries that use pan-genotypic DAAs and rarely test HCV genotype. Regarding HCV RNA, we expect more HCV RNA testing to be done in later calendar years which would confound with the calendar year variable included. Therefore, instead of including it, HCV RNA has been discussed. Furthermore, inclusion of HCV RNA in our analysis will likely show that high HCV RNA is associated with a reduced hazard for achieving SVR. Table 4 and the associated text in the methods and results sections have been revised to reflect the changes that were made. 

Comment 3. Table 2, Calendar year, “2017”-2020 should be “2018”-2020.

Response: Thank you for spotting this error, which we have now corrected.

---

## [Decision Letter · Decision Letter 1]

31 May 2023

PONE-D-23-04685R1Trends in hepatitis C virus coinfection and its cascade of care among adults living with HIV in Asia between 2010 and 2020PLOS ONE

Dear Dr. Ross,

Thank you for submitting your manuscript to PLOS ONE. After careful consideration, we feel that it has merit but does not fully meet PLOS ONE’s publication criteria as it currently stands. Therefore, we invite you to submit a revised version of the manuscript that addresses the points raised during the review process. Please ensure that your decision is justified on PLOS ONE’s publication criteria and not, for example, on novelty or perceived impact.

We look forward to receiving your revised manuscript.

Kind regards,

Chen-Hua Liu

Academic Editor

PLOS ONE

Journal Requirements:

Additional Editor Comments (if provided):

* Please make a brief discussion about the potential HCV reinfection among PLWH after achieving SVR with antiviral treatment. In Taiwan, a recent study showed that a increasing trend of HCV reinfection among HIV positive patients after the commencement of direct-acting antivirals, which may post a public health treat and should alert the physicians about the post-treatment surveillance based on public health perspectives (Liu CH, et al. Open Forum Infect Dis 2022).

Reviewers' comments:

Reviewer's Responses to Questions

**Comments to the Author**

1. If the authors have adequately addressed your comments raised in a previous round of review and you feel that this manuscript is now acceptable for publication, you may indicate that here to bypass the “Comments to the Author” section, enter your conflict of interest statement in the “Confidential to Editor” section, and submit your "Accept" recommendation.

Reviewer #1: All comments have been addressed

2. Is the manuscript technically sound, and do the data support the conclusions?

Reviewer #1: Yes

3. Has the statistical analysis been performed appropriately and rigorously? 

Reviewer #1: Yes

4. Have the authors made all data underlying the findings in their manuscript fully available?

Reviewer #1: Yes

5. Is the manuscript presented in an intelligible fashion and written in standard English?

Reviewer #1: Yes

6. Review Comments to the Author

Reviewer #1: All comments have been addressed.

7. PLOS authors have the option to publish the peer review history of their article (what does this mean?). If published, this will include your full peer review and any attached files.

Reviewer #1: No

---

## [Author Response · Author response to Decision Letter 1]

13 Jun 2023

Additional editor comments: Please make a brief discussion about the potential HCV reinfection among PLWH after achieving SVR with antiviral treatment. In Taiwan, a recent study showed that a increasing trend of HCV reinfection among HIV positive patients after the commencement of direct-acting antivirals, which may post a public health treat and should alert the physicians about the post-treatment surveillance based on public health perspectives (Liu CH, et al. Open Forum Infect Dis 2022).

Response: Thank you for the suggestion to add to the discussion the potential for HCV reinfection and the importance of post HCV treatment surveillance among those living with HIV, and for providing an associated reference. The following text was added to the discussion section: “Improved HCV screening and scale-up of HCV treatment are also important for reducing the risk of HCV re-infection amongst PLHIV (56). A recent study of HCV re-infection among an adult PLHIV cohort in Taiwan found that those living with HIV had higher HCV re-infection rates than the HIV-negative reference population, and that HCV reinfection incidence increased following the availability of DAAs in the country, further highlighting the importance of post HCV treatment surveillance among PLHIV (57).” The following references were also added: “56. Hosseini-Hooshyar S, Hajarizadeh B, Bajis S, Law M, Janjua NZ, Fierer DS, et al. Risk of hepatitis C reinfection following successful therapy among people living with HIV: a global systematic review, meta-analysis, and meta-regression. The lancet HIV. 2022;9(6):e414-e27” and “57. Liu CH, Sun HY, Peng CY, Hsieh SM, Yang SS, Kao WY, et al. Hepatitis C Virus Reinfection in People With HIV in Taiwan After Achieving Sustained Virologic Response With Antiviral Treatment: The RECUR Study. Open Forum Infect Dis. 2022;9(8):ofac348.”

---

## [Editor Report · Decision Letter 2]

15 Jun 2023

Trends in hepatitis C virus coinfection and its cascade of care among adults living with HIV in Asia between 2010 and 2020

PONE-D-23-04685R2

Dear Dr. Ross,

We’re pleased to inform you that your manuscript has been judged scientifically suitable for publication and will be formally accepted for publication once it meets all outstanding technical requirements.

Kind regards,

Chen-Hua Liu

Academic Editor

PLOS ONE

---

## [Editor Report · Acceptance letter]

21 Jun 2023

PONE-D-23-04685R2 

Trends in hepatitis C virus coinfection and its cascade of care among adults living with HIV in Asia between 2010 and 2020 

Dear Dr. Ross:

I'm pleased to inform you that your manuscript has been deemed suitable for publication in PLOS ONE. Congratulations! Your manuscript is now with our production department. 

Kind regards, 

on behalf of

Dr. Chen-Hua Liu 

Academic Editor

PLOS ONE